https://doi.org/10.1038/s42003-023-05694-1　　OPEN
# Genome-wide mapping and cryo-EM structural analyses of the overlapping tri-nucleosome composed of hexasome-hexasome-octasome moieties

Masahiro Nishimura[1,4,7], Takeru Fujii[2,7], Hiroki Tanaka[1,5], Kazumitsu Maehara [2], Ken Morishima [3], Masahiro Shimizu [3], Yuki Kobayashi[1], Kayo Nozawa[1,6], Yoshimasa Takizawa [1], Masaaki Sugiyama [3], Yasuyuki Ohkawa [2✉] & Hitoshi Kurumizaka [1✉]

The nucleosome is a fundamental unit of chromatin in which about 150 base pairs of DNA are wrapped around a histone octamer. The overlapping di-nucleosome has been proposed as a product of chromatin remodeling around the transcription start site, and previously found as a chromatin unit, in which about 250 base pairs of DNA continuously bind to the histone core composed of a hexamer and an octamer. In the present study, our genome-wide analysis of human cells suggests another higher nucleosome stacking structure, the overlapping tri-nucleosome, which wraps about 300-350 base-pairs of DNA in the region downstream of certain transcription start sites of actively transcribed genes. We determine the cryo-electron microscopy (cryo-EM) structure of the overlapping tri-nucleosome, in which three sub-nucleosome moieties, hexasome, hexasome, and octasome, are associated by short connecting DNA segments. Small angle X-ray scattering and coarse-grained molecular dynamics simulation analyses reveal that the cryo-EM structure of the overlapping tri-nucleosome may reflect its structure in solution. Our findings suggest that nucleosome stacking structures composed of hexasome and octasome moieties may be formed by nucleosome remodeling factors around transcription start sites for gene regulation.

[1] Laboratory of Chromatin Structure and Function, Institute for Quantitative Biosciences, The University of Tokyo, 1-1-1 Yayoi, Bunkyo-ku, Tokyo 113-0032, Japan. [2] Division of Transcriptomics, Medical Institute of Bioregulation, Kyushu University, 3-1-1 Maidashi, Higashi, Fukuoka 812-0054, Japan. [3] Institute for Integrated Radiation and Nuclear Science, Kyoto University, Kumatori, Sennan-gun, Osaka 590-0494, Japan. [4] Present address: Epigenetics and Stem Cell Biology Laboratory, National Institute of Environmental Health Sciences, 111 TW, Alexander Drive, Research Triangle Park, NC 27707, USA. [5] Present address: Department of Structural Virology, National Center for Global Health and Medicine, 1-21-1 Toyama, Shinjuku-ku, Tokyo 162-8655, Japan. [6] Present address: School of Life Science and Technology, Tokyo Institute of Technology, 4259 Nagatsuta-cho, Midori-ku, Yokohama, Kanagawa 226-8501, Japan. [7] These authors contributed equally: Masahiro Nishimura, Takeru Fujii. ✉email: yohkawa@bioreg.kyushu-u.ac.jp; kurumizaka@iqb.u-tokyo.ac.jp

Genetic information is encoded in DNA, which is compactly accommodated as chromatin within the nucleus in eukaryotic cells[1]. In chromatin, four core histone proteins, H2A, H2B, H3, and H4, form a histone octamer, in which two H2A–H2B and H3–H4 heterodimers associate with each other, and wrap 145–147 base pairs (bps) of DNA around the lateral periphery of the histone octamer[2]. The resulting histone–DNA complex is termed the nucleosome core particle (NCP) and is a fundamental structural unit of chromatin. The NCPs are connected by linker DNA segments and form fibers with a bead-on-a-string appearance[3]. The bead-on-a-string fiber may be further folded into an unknown higher-order chromatin architecture in the nucleus[4].

The DNA wrapped in the nucleosome generally becomes inaccessible to DNA-binding proteins, which function in genomic DNA regulation[5–7]. To overcome the nucleosome barrier during the replication, repair, recombination, and transcription processes, nucleosomes are dissociated from and/or slide along the DNA during the nucleosome remodeling process, which is mediated by a group of enzymes called nucleosome remodelers[8,9]. In particular, during the transcription initiation process, the nucleosome located at the transcription start site (TSS) of a gene could be removed by the nucleosome remodeling activity for the subsequent assembly of the transcription machinery, such as RNA polymerase II[10,11]. In this process, the nucleosome remodeling potentially induces the collision of neighboring nucleosomes and forms the overlapping di-nucleosome (OLDN), in which the hexameric nucleosome (hexasome) lacking one H2A–H2B dimer associates with the nucleosome (octasome)[12–14].

In the crystal structure of the OLDN, ~250 bp DNA is continuously wrapped in three turns around the histone core[14]. Consistent with the OLDN structure, the OLDN formation protects ~250 bp DNA from digestion by micrococcal nuclease (MNase), which preferentially degrades nucleosome-free DNA regions[14]. A genome-wide analysis revealed that the protection of 250 bp from MNase is predominantly observed at the downstream regions immediately adjacent to transcription start sites (TSSs) in human cells[14]. This suggests that the OLDN formation may be mediated by nucleosome remodelers, which function around the TSS for transcription initiation[14].

In the present study, we found a higher-order nucleosome stacking structure, possibly corresponding to the overlapping tri-nucleosome (OLTN), which may be formed at the downstream regions of TSSs in the human genome. We then determined the structure of the OLTN, composed of hexasome, hexasome, and octasome, by cryo-electron microscopy (cryo-EM). The cryo-EM structure of the OLTN may reflect its structure in solution as revealed by small-angle X-ray scattering and coarse-grained molecular dynamics simulation analyses.

## Results

### The overlapping tri-nucleosome may be formed downstream of transcription start sites in the genome. A previous genome-wide MNase analysis revealed that approximately 250-bp, corresponding to the OLDN, are protected just downstream of TSSs in human cells[14]. This suggested that nucleosome collisions, probably by the nucleosome remodeling activity, may promote the OLDN formation around the TSS regions in the genome[14]. If the nucleosome remodeling constantly occurs with the OLDN around TSS regions, then another nucleosome could collide with the OLDN and form the OLTN, a further nucleosome stacking structure with three nucleosomes.

To test this possibility, we first performed the MNase protection analysis with chromatin isolated from HeLa cells. The isolated chromatin was extensively digested by MNase until genomic DNA fragments larger than 150 bp were barely detected (Fig. 1a and Supplementary Fig. 1a). Since the extensive MNase digestion generally removes DNA-binding proteins, this step is expected to reduce the possible contamination from DNA fragments protected by DNA-binding proteins other than histones. We then prepared the resulting MNase-treated 200–300 and 300–400 bp genomic DNA fragments, which may correspond to OLDN and OLTN, respectively, by polyacrylamide gel electrophoresis, and performed a next-generation sequencing (NGS) analysis. Consistent with the previous NGS analysis, we confirmed that the 250 bp fragments corresponding to the OLDN were accumulated in the regions downstream of TSSs in HeLa cells (Fig. 1b, c and Supplementary Fig. 1b, c). Interestingly, we found that 300–350 bp DNA fragments, which may correspond to the OLTN, were also accumulated in this region (Fig. 1d, e and Supplementary Fig. 1d–f). This suggested that OLTNs, in addition to OLDNs, may form in the downstream regions of TSSs in human cells.

### Cryo-EM visualization of the OLTN. We then tested the OLTN formation in vitro, using a 353 bp DNA fragment containing 125, 103, and 125 bp DNA segments of the Widom 601 nucleosome positioning sequence, for in vitro nucleosome reconstitution (Fig. 2). In the previous OLDN reconstitution, the 250 bp tandem Widom 601 DNA was used as the template DNA, in which the 44 bp connection site region (22 bp of downstream and upstream regions of each Widom 601 DNA), was deleted[14]. In the present OLTN reconstitution, the 44 bp DNA region of the second connection site was also deleted (Fig. 2a). Purified human histones, H2A, H2B, H3, and H4, were mixed with the 353 bp DNA fragment, and the histone complex was assembled on the DNA by the salt dialysis method[15]. The reconstituted histone–DNA complex was purified by native polyacrylamide gel electrophoresis (Fig. 2b, c, and Supplementary Fig. 2). The additional bands just after the OLTN reconstitution may be the OLDN (migrating faster than the OLTN), the OLTN with improper additional histone binding (migrating slower than the OLTN), and the OLTN with the different positioning of subnucleosome moieties (Supplementary Fig. 2, Input). The reconstituted OLTN sample was subjected to MNase digestion coupled with NGS analysis. We found that 335–338 bp DNA fragments were protected from the MNase digestion (Fig. 2d). This is consistent with the MNase-seq analysis data, in which 300–350 bp DNA fragments, probably corresponding to the OLTN, are protected from MNase digestion in the genomic DNA (Fig. 1d, e).

The reconstituted OLTN was fractionated by sucrose gradient ultracentrifugation with glutaraldehyde fixation (GraFix) and subjected to cryo-EM data collection with a 300 kV electron microscope. From the 6953 electron micrographs, 1.7 million particles were identified (Supplementary Fig. 3). We then obtained the cryo-EM structure of the OLTN (Fig. 3a), with an overall resolution of 7.6 Å. In the OLTN structure, another hexasome is associated with the OLDN, and consequently, the hexasome (proximal), hexasome (central), and octasome (distal) moieties are aligned in this order (Fig. 3b–d). A 3D map (8.5%) was obtained with a possibly different orientation between the OLDN part and the proximal hexasome portion, but this is uncertain due to the noisy background (Supplementary Fig. 3). We refined each nucleosome moiety, proximal hexasome (5.2 Å), central hexasome (5.9 Å), and distal octasome (4.5 Å), and obtained a composite map (Supplementary Fig. 3b).

To determine whether the OLTN formation occurs with a native DNA sequence, we repeated the OLTN reconstitution experiment with the 350 bp DNA fragment containing the human genome sequence identified as the SMARCC1 and SMARCC2

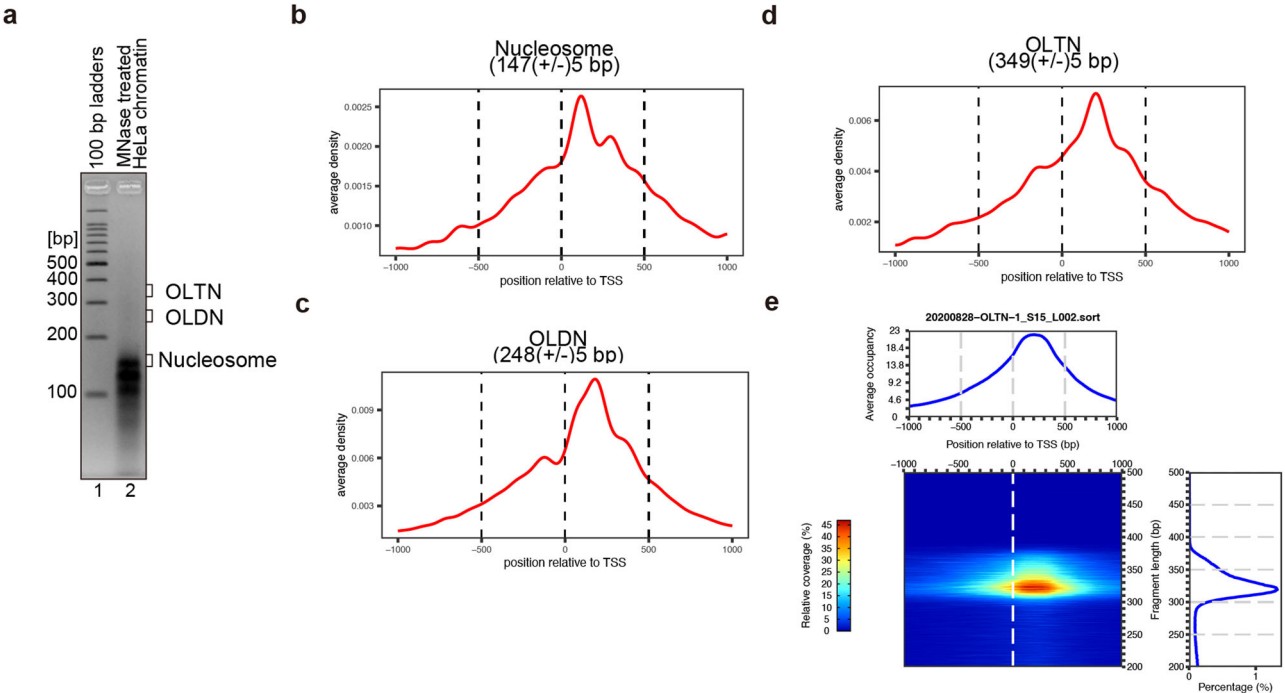

**Fig. 1 MNase-seq analyses of the HeLa cell chromatin. a** MNase treatment of chromatin isolated from HeLa cells. MNase (3 units/μL) was used to digest chromatin from HeLa cells. The resulting genomic DNA fragments protected from MNase digestion by the formation of the nucleosome, overlapping di-nucleosome (OLDN), and overlapping tri-nucleosome (OLTN) were analyzed by agarose gel electrophoresis with ethidium bromide staining. The DNA fragments with lengths of about 140–150, 200–300, and 300–400 bp, which may correspond to nucleosomes, OLDNs, and OLTNs, respectively, were extracted from the agarose gel and subjected to next-generation sequencing analysis. **b–d** Aggregation plots of the DNA fragments corresponding to nucleosomes (**b**), OLDNs (**c**), and OLTNs (**d**) relative to TSSs. The reads with the indicated lengths extracted from the sequencing data were mapped between a ±1 kb range around TSSs. **e** Heatmap representation of the aggregation plot analysis of the overlapping tri-nucleosome, expanded by the DNA fragment length. The histogram of the DNA fragment lengths is presented in the right panel.

accumulated region (conservative IDR thresholded ChIP-seq peaks; ENCSR000EDM, ENCSR000EDL). SMARCC1 and SMARCC2 are subunits of a prominent nucleosome remodeler, the mammalian SWI/SNF complex, which may promote the OLDN formation[12]. We selected the 350 bp DNA fragment with the DNA sequence ~70 bp downstream from the TSS. By 2D class averages, we then confirmed that the OLTN can be formed with this genomic DNA sequence (Supplementary Fig. 4).

**The OLTN structure**. In the OLTN structure, a 349 bp DNA segment is continuously wrapped in the hexasome–hexasome–octasome core and adopts a "slinky"-like shape (Fig. 4a). In the distal octasome part, 145 bp of DNA contact with the histone octamer (Fig. 4a), while in the proximal and central hexasomes, 97 bp of DNA directly bind to the histone surface (Fig. 4a). The hexasome–hexasome and hexasome–octasome boundaries are connected by 7 and 3 bp DNA segments, respectively (Fig. 4a). The 97 and 145 bp DNA regions wrapped around the histone hexamer and octamer, respectively, are aligned horizontally in the OLTN structure (Fig. 4a). The H2A–H2B dimers are missing at the hexasome–hexasome and hexasome–octasome interfaces (Fig. 4b–d). This may reduce steric hindrance at the interfaces between hexasome and hexasome or hexasome and octasome, and allow the horizontal DNA alignment in the OLTN (Fig. 4a).

**Small angle X-ray scattering and molecular dynamics simulation analyses of the OLTN in solution**. To acquire structural insights into the structure of the OLTN in solution, we conducted a small angle X-ray scattering coupled with analytical ultra-centrifugation (AUC-SAXS) analysis[16,17]. In this method, the bona fide SAXS of OLTN can be extracted from the scattering of

the sample solution containing OLTN and aggregates. The OLTN was reconstituted with the same 353 bp DNA fragment (Widom 601 sequence) used in the cryo-EM analysis, and the sample was subjected to the SAXS experiment without crosslinking. The SAXS curve of the OLTN is quite similar to that obtained from the cryo-EM structure of the OLTN (Fig. 5a). Furthermore, we constructed a dummy atom model of the OLTN based on the SAXS, for comparison with the cryo-EM structure. It fits very well with the dummy atom model, suggesting that the cryo-EM structure likely represents a major form of the OLTN in solution (Fig. 5b). It should be noted that the SAXS curve of the OLTN slightly deviates from that calculated from the cryo-EM structure (Fig. 5a). This may reflect the structural dynamics of the OLTN in solution. To address this, we next performed a coarse-grained molecular dynamics (CG-MD) simulation analysis, which revealed that the trajectory of the OLTN structures is consistent with the SAXS and cryo-EM analyses (Fig. 5a). The representative structure in the CG-MD trajectory is shown in Fig. 5c. Given the dynamic arrangement of the hexasomal and octasomal portions in the OLTN, the histone–DNA and histone–histone interactions between the subnucleosomal portions may not be stable but transient around the cryo-EM structure.

## Discussion

We previously reported the structure of the OLDN, composed of a hexasome and an octasome, and its presence in the regions immediately downstream of TSSs in human cells[14]. Consistently, the OLDN was reportedly visualized by cryo-electron microscopy in isolated human mitotic chromosomes[18]. These facts imply that the OLDN may be persistently maintained throughout the cell cycle. In the present study, our genome-wide MNase-seq analysis

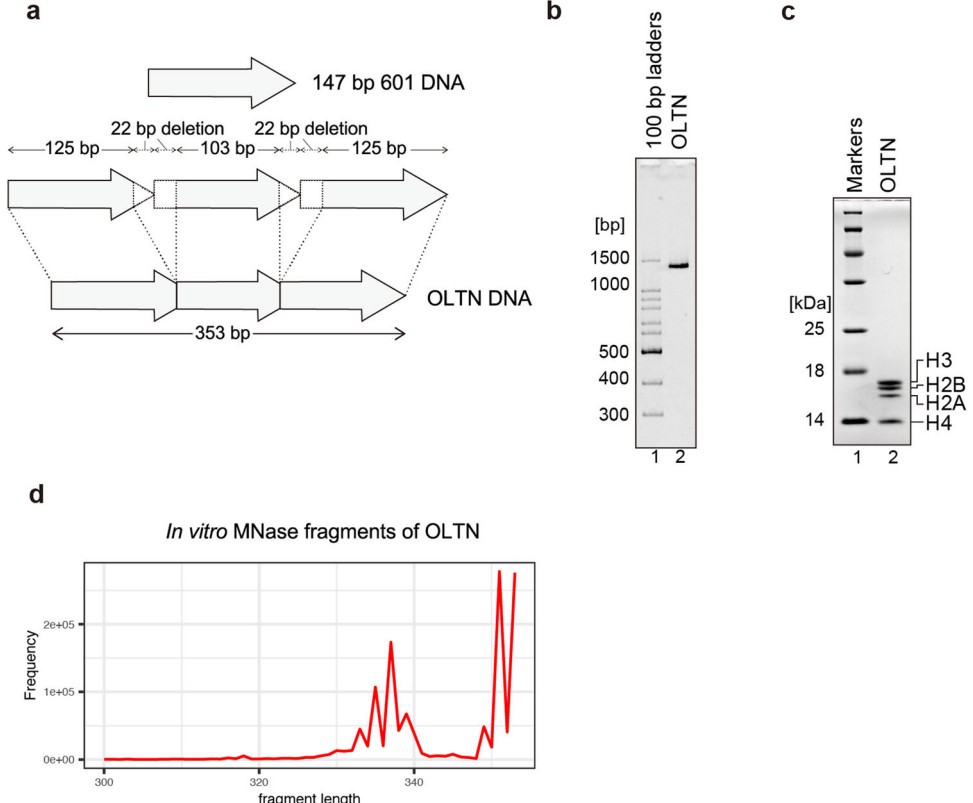

**Fig. 2 Reconstitution of the overlapping tri-nucleosome. a** Schematic representation of the 353 bp DNA substrate used for the reconstitution of the OLTN (OLTN DNA). The OLTN DNA is derived from the three tandemly connected Widom 601 sequences, represented by arrows, and the 22 bp segments located at the junctions are deleted. **b** The purified OLTN was analyzed by non-denaturing polyacrylamide gel electrophoresis with ethidium bromide staining. **c** The histone content of the purified OLTN was analyzed by SDS–PAGE with Coomassie Brilliant Blue staining. **d** Histogram of the DNA lengths obtained by the in vitro MNase treatment assay with purified OLTN. The DNA fragments obtained after MNase treatment were analyzed by massively parallel paired-end sequencing. The 5′-ends of the DNA fragments are mapped on the substrate DNA sequence, and the fragment lengths are estimated.

revealed that OLTNs may also form in the downstream regions of TSSs (Fig. 1). We successfully reconstituted and determined the cryo-EM structure of the OLTN, in which three nucleosomes, hexasome, hexasome, and octasome, are associated with short linker DNA segments (Figs. 3 and 4). It should be noted that, in the current method, the possible existence of the OLDN and OLTN can be deduced in cells; however, their prevalence relative to the normal nucleosome is currently difficult to evaluate. Therefore, the rates of OLDN and OLTN formation relative to the nucleosome may be an important future issue to solve.

Considering its formation in the downstream region of the TSS, the OLTN may be formed by the collision of an additional nucleosome with the OLDN, as a consequence of the nucleosome remodeling activity. As another possibility, the OLDN and OLTN may also be formed through nucleosome transfer during transcription elongation by RNA polymerase II[19,20]. In this process, the nucleosome located in the DNA region downstream of RNA polymerase II is disassembled and transferred to the upstream transcribed DNA region[20]. This nucleosome transfer may induce the OLDN and OLTN formation. RNA polymerase II-mediated nucleosome disassembly and reassembly may also function in resolving the OLDN and OLTN into the canonical nucleosome. The assembly and disassembly processes of the OLDN and OLTN remain to be clarified.

The first nucleosome (+1 nucleosome) of the protein-coding regions of genes is reportedly positioned approximately 40 bp downstream from the TSS[21–23], and the preinitiation complex (PIC) containing RNA polymerase II is bound to the +1 nucleosome[24,25]. To do so, the nucleosome-depleted region

(NDR) must be created by nucleosome remodelers before the PIC assembly around the TSS[26]. The SWI/SNF nucleosome remodeling complex, which promotes nucleosome sliding, is a possible candidate as a nucleosome remodeler to produce the NDR by its nucleosome sliding activity[27–29]. The nucleosome remodeling may induce nucleosome collision, forming the OLDN and OLTN in the region downstream of the TSS[12,13]. The cryo-EM structures of the mammalian SWI/SNF family remodelers, the human BAF and PBAF complexes, bound to the nucleosome have been reported[30,31]. In these structures, the nucleosome is located on the solvent-accessible surface and can associate with another nucleosome particle without serious steric hindrance[30,31]. Therefore, the OLDN and OLTN could bind to the BAF and PBAF complexes by a mechanism similar to that of the nucleosome, and become substrates for further nucleosome remodeling.

The +1 nucleosome may function as a barrier for transcription elongation[32–35], and the OLDN and OLTN formed at the +1 position may enhance this function of the +1 nucleosome. In the PIC structures, the nucleosomes are quite flexible and largely exposed to the solvent[25]. The OLDN and OLTN can be accommodated within the PIC without substantial steric clashes. Further studies are awaited to clarify the relationship between OLDN/OLTN formation and transcription status in genes.

In the previous crystal structure of the OLDN, the hexasome–octasome moieties are more compactly associated than those in the OLTN structure (Supplementary Fig. 5). This may be due to restriction of the OLDN dynamics by crystal packing effects, or it may reflect one of the possible arrangements of the hexasome–octasome moieties. The hexasome–octasome

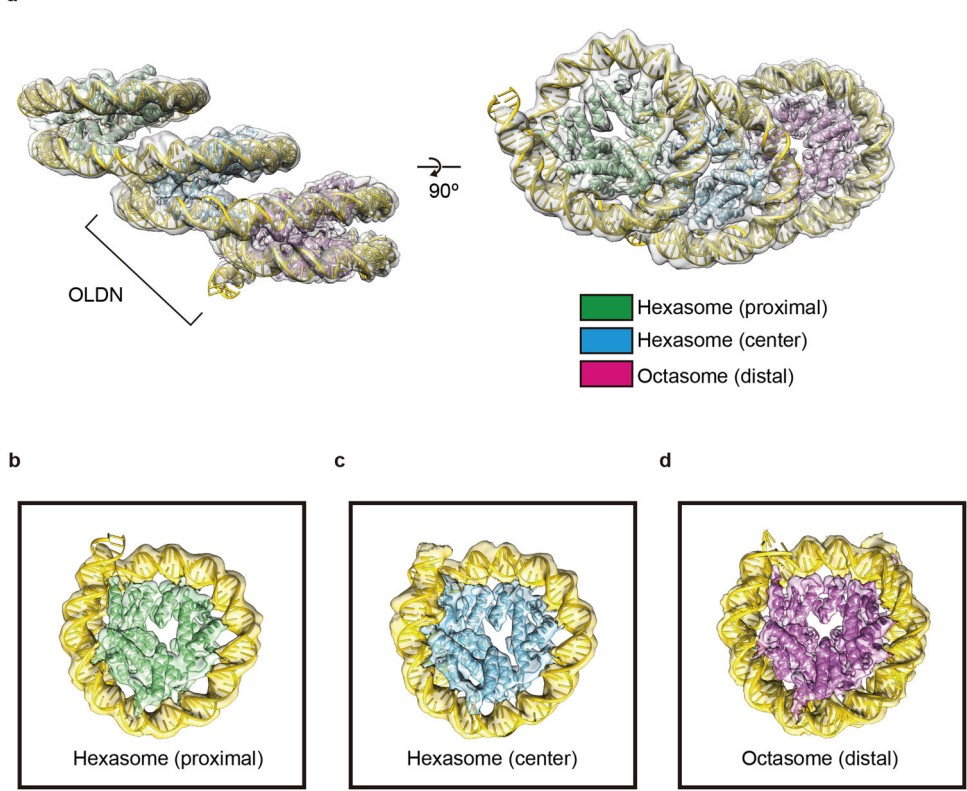

**Fig. 3 Cryo-EM structure of the OLTN. a** Cryo-EM structure of the OLTN. The proximal hexasome, center hexasome, and octasome moieties are shown in green, blue, and magenta, respectively. **b–d** Structures of the proximal hexasome, the central hexasome, and the distal octasome, embedded in the corresponding cryo-EM densities generated by the multibody refinement.

arrangement of the OLTN is consistent with the structural range of the OLDN in solution[36]. These findings suggest that the octasome and hexasome moieties in the OLTN, as well as the OLDN, may have dynamic properties that allow regulation of transcription through the +1 nucleosome downstream of the TSS. To address this fundamental question, studies of OLTN and OLDN transcription by RNA polymerase II are awaited.

## Methods

**Preparation of DNA fragments for next-generation sequencing analyses.** The HeLa nuclei were prepared as previously described[14]. The HeLa cells ($5 \times 10^7$) were suspended in 15 ml of washing solution, containing 15 mM Tris–HCl (pH 8.0), 15 mM NaCl, 60 mM KCl, 300 mM sucrose, and a protease inhibitor (Roche). The cell suspension was mixed with 15 ml of detergent buffer, containing 15 mM Tris–HCl (pH 8.0), 15 mM NaCl, 60 mM KCl, 300 mM sucrose, protease inhibitor (Roche), and 1% NP-40, was then added to the cell suspension, and the mixture was rotated at 4 °C for 10 minutes. The HeLa nuclei were collected by centrifugation and resuspended in 100 µL of the washing solution. A large amount of MNase (3 units/µL) and $CaCl_2$ (2 mM) were added to the HeLa nuclei suspension. The MNase treatment was performed at 37 °C for 30 min and then terminated by EDTA (pH 8.0) addition at a final concentration of 50 mM. The resulting reaction mixture was collected by centrifugation, and the supernatant was deproteinated by adding a mixture of proteinase K (0.9 mg/ml, Roche) and 0.5% SDS, followed by phenol/chloroform extraction. The MNase fragments were precipitated by ethanol and electrophoresed in a 4% agarose gel. DNA samples with sizes of 140–150, 200–300, and 300–400 bp were extracted from the agarose gel and used for the NGS analyses.

**Next-generation sequencing analyses.** The paired-end library of MNase-seq was prepared with a ThruPLEX® DNA-Seq Kit (Clontech, #R400674), and sequencing was performed using an Illumina NovaSeq6000 sequencer (in vivo data) and an Illumina MiSeq sequencer (in vitro data). The reads were mapped to the human genome (GRCh38) using Bowtie2[37] (version 2.3.5.1) with default parameters. BAM files of in vivo assays were created using SAMtools[38] (version 1.9). BAM files were input to plot2DO[39], to create heatmaps. To create aggregation plots, the mapped reads with presumable lengths of nucleosomal DNA were extracted from each BAM file using SAMtools[38] OLTN: 337 (±5) bp, OLDN: 248 (±5) bp, and mono-nucleosome: 147 (±5) bp. The resulting files of the extracted reads were input to agplus[40] to create the aggregation plots.

**Reconstitution and purification of the OLTN.** The OLTN was prepared with the 353 bp DNA fragment derived from the Widom 601 sequence (OLTN DNA) and a purified histone octamer. The OLTN DNA fragments were tandemly inserted into the pGEM-T Easy vector (Promega), and the plasmid was amplified in DH10alpha cells. The OLTN DNA fragment was then isolated from the vector by EcoRV restriction enzyme (Takara) treatment, and purified by polyethylene glycol precipitation and TSKgel DEAE-5PW (TOSOH) anion exchange column chromatography. The histone octamer was prepared from recombinant human histones H2A, H2B, H3, and H4, as previously described[15]. In this study, we employed H3.1 as the histone H3. In brief, the four histones were mixed at an equal molar ratio in denaturing buffer, containing 20 mM Tris–HCl buffer (pH 7.5), 7 M guanidine hydrochloride, and 20 mM 2-mercaptoethanol, and the mixture was dialyzed against refolding buffer, containing 10 mM Tris–HCl (pH 7.5), 2 M NaCl, 1 mM

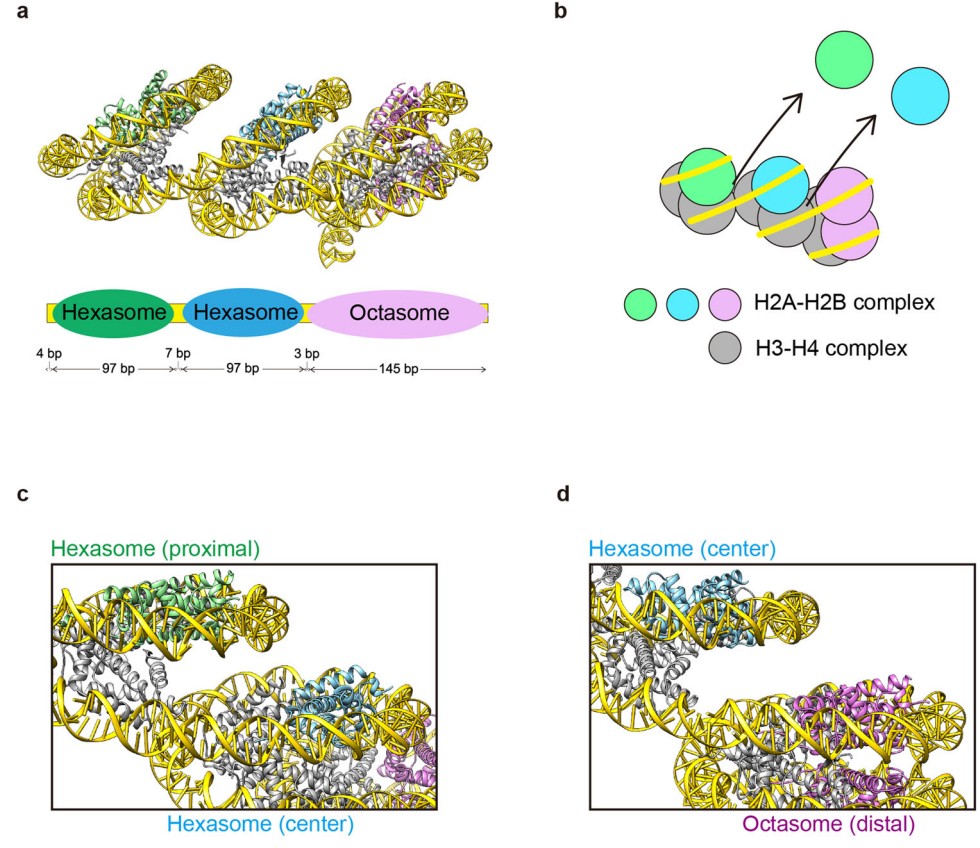

**Fig. 4 Internal organization of the OLTN. a** Schematic representation of the DNA regions occupied by the proximal hexasome, the central hexasome, and the distal octasome in the OLTN structure. **b** Illustration of the OLTN structure. The H2A–H2B complexes are colored green, blue, and magenta. The H3–H4 complexes are gray. **c** and **d** Close-up views of the hexasome–hexasome interface (**c**) and the hexasome–octasome interface (**d**).

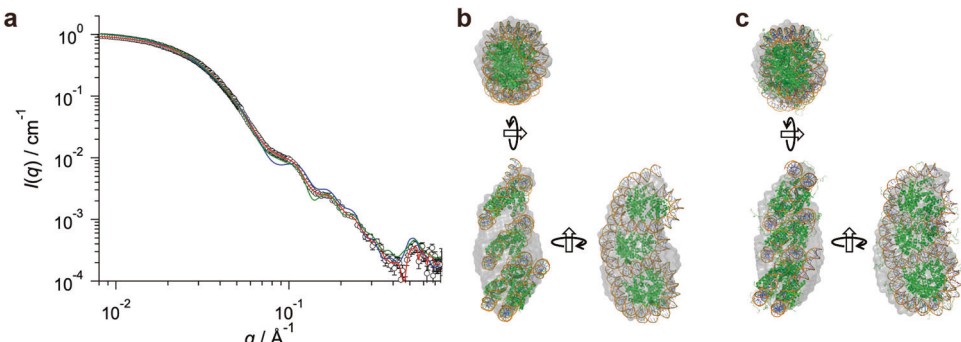

**Fig. 5 SAXS and 3D structures of the OLTN. a** Experimental SAXS curve subjected AUC-SAXS treatment (open circles), and SAXS curves computed from the cryo-EM model (blue line), ab initio model (red line), and CG-MD model (green line). **b** and **c** Superpositions of the ab initio model (gray envelope) and cryo-EM model (ribbon diagram, **b**) and CG-MD model (ribbon diagram, **c**).

EDTA, and 5 mM 2-mercaptoethanol. After the dialysis, the refolded histone octamer was purified by size exclusion column chromatography in the refolding buffer.

The OLTN was reconstituted by the salt dialysis method, as previously described[15]. The purified histone octamer and the DNA were mixed at a 3:1 molar stoichiometry in reconstitution buffer-high, containing 10 mM Tris–HCl (pH 7.5), 2 M KCl, 1 mM EDTA (pH 8.0), and 1 mM dithiothreitol. The KCl concentration was decreased to 250 mM by continuous buffer exchange with peristaltic pumps. The reconstituted sample was fractionated by non-denaturing gel electrophoresis using a Prep

Cell apparatus (Bio-Rad) in the elution buffer, containing 20 mM Tris–HCl (pH 7.5) and 1 mM dithiothreitol. The peak fractions were collected and concentrated by filtration. The concentration of OLTN was determined by the absorbance at 260 nm.

The sequence of the OLTN DNA is as follows: 5′-ATCGAGA ATCCCGGTGCCGAGGCCGCTCAATTGGTCGTAGACAGCT CTAGCACCGCTTAAACGCACGTACGCGCTGTCCCCCGCG TTTTAACCGCCAAGGGGATTACTCCCTAGTCTCCAGGCT CGAGCTCAATTGGTCGTAGACAGCTCTAGCACCGCTTAA ACGCACGTACGCGCTGTCCCCCGCGTTTTAACCGCCAAG GGGATTACTCCCTAGTCTCCAGGCTCGAGCTCAATTGGT

CGTAGACAGCTCTAGCACCGCTTAAACGCACGTACGCG
CTGTCCCCCGCGTTTTAACCGCCAAGGGGATTACTCCCT
AGTCTCCAGGCACGTGTCAGATATATACATCCGAT-3′.

For the OLTN containing the native DNA sequence, the purified histone octamer and the native DNA (350 bp) were mixed at a 4.8:1 molar stoichiometry in reconstitution buffer-high. The following analyses of the OLTN containing native DNA sequence were performed in the same manner as for the OLTN.

The sequence of the native OLTN DNA (S100A13 gene locus, ENSEMBL_ID: ENSG00000189171) is as follows: 5′-TTTGAA CCCAGTCCAATAAAACCTCAAAACCGGTGCACTTTCTAC CGTATCCTGAGGCTTCTTTACTTTGGGGCCCTGGTTAGC CTTAGCAGCCTAGTTTTCTACATCCTTCATGCCAGTTGA ATGAAACTACTGACATGCTCCACATTCTTCTTTCTCCCAT GCTTTTGTTCATTCAGTTACCTCCTCCTAAAATGTCTGCA TTTACCCAGATAATCTTCCAATGGAAATCCATGGTTCAA GTGCCACCTCTTCAGGAAAGCCATCTGACTTCAATCAGG TTAATAATCTCTCATAACCCTTTCTGGTACATCTGTTAAG GCTCTGTGTATTTCCTGGAGTTCATGGTT-3′.

**MNase treatment of the purified OLTN**. The purified OLTN (0.2 μM) was incubated with 25 mU/μL MNase (Takara) at 37 °C for 9 minutes in 296 μL of reaction solution, containing 41 mM Tris–HCl (pH 7.5 and 8.0), 5 mM NaCl, 2.5 mM CaCl$_2$, 5% glycerol, 1 mM 2-mercaptoethanol, and 0.8 mM dithiothreitol. The MNase reaction was terminated by adding an equal volume of ProK solution, containing 200 mM Tris–HCl (pH 8.0), 80 mM EDTA (pH 8.0), 0.5 mg/ml proteinase K (Roche), and 0.25% SDS. The resulting DNA fragments were collected by phenol–chloroform–isoamyl alcohol extraction followed by ethanol precipitation and then electrophoresed in an 8% urea gel. The DNA sample with fragment sizes of 330–350 bp was extracted from the gel by electroelution and used for the NGS analysis.

**Preparation of the OLTN for cryo-EM analysis**. The OLTN and the OLTN containing the native DNA sequence for the cryo-EM analysis were prepared by the gradient fixation method (GraFix)[41]. The gradient solution contained 20 mM HEPES–KOH (pH 7.5), 50 mM KOAc, 0.2 μM Zn(OAc)$_2$, 0.1 mM TCEP (pH 8.0), 10–25% (w/v) sucrose, and 0–0.1% (v/v) glutaraldehyde. The reconstituted OLTN was purified by non-denaturing polyacrylamide gel electrophoresis using a Prep Cell apparatus (Bio-Rad), concentrated by filtration, applied to the top of the gradient solution, and centrifuged at 27,000 rpm for 16 h (Beckman Coulter Optima™; SW41 Ti rotor). After centrifugation, aliquots were carefully collected from the top of the gradient. The peak fractions were analyzed by non-denaturing gel electrophoresis and then desalted using a PD-10 column (Cytiva) in a final buffer, containing 20 mM HEPES–KOH (pH 7.5), 50 mM KOAc, and 2 mM TCEP (pH 8.0). The fixed OLTN and the OLTN containing the native DNA sequence were concentrated by filtration and stored on ice.

**Cryo-EM data collection**. For the OLTN, the Quantifoil grids (R1.2/1.3 200-mesh copper) were treated with ethyl acetate and glow-discharged before use. Aliquots of the OLTN (0.3 mg/ml) were blotted in a Vitrobot Mark IV (Thermo Fisher Scientific) at blot force 5 for 10 s, under 100% humidity at 16 °C, and the grids were immediately plunged into liquid ethane to embed the sample particles into amorphous ice. Data collections were performed on a Krios G3i cryo-transmission electron microscope (Thermo Fisher Scientific) operated at 300 kV. The micrographs were automatically recorded on a K3 BioQuantum (Gatan) direct electron detector, calibrated at a pixel size of 1.07 Å in the

electron counting mode, using a slit width of 25 eV and retaining 40 frames with a total dose of 55.1 electron/Å, using the EPU software. For the OLTN containing the native DNA sequence, the Quantifoil grids (R1.2/1.3 200-mesh copper) were treated with ethyl acetate and glow-discharged before use. Aliquots of the OLTN containing the native DNA sequence (1 mg/ml) were blotted in a Vitrobot Mark IV (Thermo Fisher Scientific) at blot force 0 for 6 s, under 100% humidity at 16 °C, and the grids were immediately plunged into liquid ethane to embed the sample particles into amorphous ice. Data collections were performed on a Krios G4 cryo-transmission electron microscope (Thermo Fisher Scientific) operated at 300 kV. The micrographs were automatically recorded on a K3 BioQuantum (Gatan) direct electron detector, calibrated at a pixel size of 1.06 Å in the electron counting mode, using a slit width of 20 eV and retaining 40 frames with a total dose of 62.0 electron/Å, using the EPU software. The defocus range was between 1 and 2.5 μm. The recording parameters are summarized in Table 1.

**Image processing**. For both the OLTN and the OLTN containing the native DNA sequence, the movie frames were aligned by the MotionCor2 software[42], and the contrast transfer functions for each frame were estimated by the CTFFIND4 software[43]. The following image processing of the OLTN structure was performed using the RELION 3.1 software[44]. In total, 1,718,300 particles from 6953 electron micrographs were automatically picked using the Laplacian-of-Gaussian filter, with a box size of 300 pixels. Subsequently, the junk particles were removed through one round of 2D classification. The initial 3D model was then generated by

| **Table 1 Cryo-EM data collection and validation.** | |
|---|---|
| | **Overlapping tri-nucleosome (PDB: 8IHL, EMDB: EMD-35448)** |
| *Data collection and processing* | |
| Microscopy | Krios G3i |
| Detector | K3 BioQuantum |
| Magnification | 81,000 |
| Voltage (kV) | 300 |
| Electron exposure (e−/Å²) | 55.1 |
| Defocus range (μm) | 1–2.5 |
| Pixel size (Å) | 1.07 |
| Symmetry imposed | C1 |
| Initial particle (no.) | 1,718,300 |
| Final particle (no.) | 148,082 |
| Map resolution (Å) (FSC = 0.143) | 7.6 |
| Map resolution range (Å) | 5.8–9.9 |
| Map sharpening B factor (Å²) | −364.5 |
| *Refinement* | |
| Initial model used (PDB code) | 5GSE |
| *Model composition* | |
| Protein residues | 1728 |
| Nucleotide residues | 706 |
| *R.m.s. deviations* | |
| Bond lengths (Å) | 0.004 |
| Bond angles (degree) | 0.803 |
| MolProbity score | 1.75 |
| Clashscore | 17.76 |
| Poor rotamers (%) | 0.00 |
| *Ramachandran plot* | |
| Favored (%) | 99.17 |
| Allowed (%) | 0.83 |
| Disallowed (%) | 0 |
| Q-score (all chains) | 0.109 |

de novo modeling for the reference model of the first 3D classification. After two rounds of 3D classification, the selected OLTN structure, in which each proximal hexasome, center hexasome, and octasome were clearly visible, and comprising 148,082 particles, was refined using a global soft mask followed by post-processing. The resolution for the final map was estimated as 7.6 Å based on the Fourier shell correlation criteria of 0.143. To resolve the octasome unit and the two hexasome units, multi-body refinements[45] using each mask were performed and followed by the postprocesses. The final resolutions of the proximal hexasome, center hexasome, and octasome were 5.2, 5.9, and 4.5 Å, respectively.

The following image processing of the OLTN containing the native DNA sequence was performed using the RELION 4.0 software[46]. In total, 1,666,759 particles from 4851 electron micrographs were picked automatically, followed by 2D classification.

**Model building and refinement**. The structural model was built from segments of the octasome unit and the hexasome unit extracted from the crystal structure of the OLDN (PDB ID: 5GSE)[14]. The octasome unit and the two hexasome units were fitted into the corresponding cryo-EM densities using rigid-body fitting in UCSF Chimera[47]. The DNA segments between each unit were connected and some histone residues outside the cryo-EM density were removed manually, using the Coot software[48,49]. The model was refined by phenix_real_space_refine[50] using the histone octamer segment of the nucleosome structure (PDB ID: 5Y0C)[51] as the reference model. The DNA sequence of the OLTN was estimated from the sequencing analysis of the MNase fragment of the purified OLTN. The model parameters calculated by the MolProbity software[52] and the Q-score[53] obtained from wwPDB validation are summarized in Table 1.

**Integrated method of analytical ultracentrifugation (AUC) and small-angle X-ray scattering (SAXS): AUC–SAXS**
*Sample*. A 2.9 mg/mL OLTN solution, in buffer containing 20 mM Tris–HCl (pH 8.0), 50 mM NaCl, and 1 mM dithiothreitol, was utilized.

*SAXS measurement*. SAXS measurement was performed using a NANOPIX (Rigaku, Japan) equipped with a point-focused generator of a Cu-Kα source, MicroMAX-007 HFMR (Rigaku, Japan) (wavelength = 1.54 Å). Scattered X-rays were counted using a HyPix-6000 detector (Rigaku, Japan). The sample-to-detector distances were set at 1330 and 350 mm, covering a $q$-range of 0.008–0.750 Å$^{-1}$ ($q$: magnitude of scattering vector). All measurements were performed at 25 °C. The two-dimensional scattering patterns were converted into one-dimensional scattering profiles with the SAngler software (version 2.1.45)[54]. After correcting for transmission and subtracting the buffer scattering, we obtained the absolute scattering intensity by using the standard scattering intensity of water ($1.632 \times 10^{-2}$ cm$^{-1}$)[55].

*AUC measurement*. To investigate the potential presence of aggregated components in the sample solution, AUC measurements were performed using a ProteomeLab XL-I instrument (Beckman Coulter, USA), with Rayleigh interference optics at 25 °C and a rotor speed of 30,000 rpm. The weight fraction distribution as a function of the sedimentation coefficient, normalized to the value at 20 °C in pure water, and the frictional ratio was calculated using the SEDFIT software (version 15.01c)[56]. The molecular weights of the components were determined based on the sedimentation coefficient and frictional ratio[57]. From the AUC data, the presence of OLTN and three aggregates in the

sample solution was observed as follows: Component 1 (OLTN) {466 kDa, 64.3%}, Component 2 (Aggregate 1) {731 kDa, 13.2%}, Component 3 (Aggregate 2) {882 kDa, 16.4%}, and Component 4 (Aggregate 3) {1150 kDa, 6.1%} {obtained molecular weights, weight fractions}.

*AUC–SAXS*. AUC–SAXS treatment[16,17] was applied to extract the scattering profile of OLTN from the measured SAXS data, including the aggregates. As a result, the gyration radius, determined through Guinier analysis, decreased from $70.1 \pm 0.4$ Å (non-treated experimental data) to $61.0 \pm 0.4$ Å (AUC–SAXS-treated data), indicating the true gyration radius of OLTN in solution.

**3D-modeling from the SAXS data**
*Ab initio 3D modeling*. An ab initio model of OLTN in solution was built with the dummy atom method, DAMMIF, in the ATSAS software package (version 3.1.3)[58], based on the SAXS profile.

*Coarse-grained molecular dynamics (CG-MD) modeling*. The possible structures of OLTN were explored through CG-MD simulations, using CafeMol[59]. The OLTN model with full-length histone proteins was prepared in two steps. First, the missing residues in the histone cores were filled in by using the known 3D structure (PDB:5GSE)[14]. Second, histone tails were generated using PyMOL[60]. The 3SPN.2 DNA model, AICG2+ model, and a flexible local potential were employed for DNA, histone cores, and histone tails, respectively[61–63]. Charges on the histone cores were defined using RESPAC[64]. For the histone tails, +1 charge was assigned to each arginine or lysine bead, and −1 charge was assigned to each aspartic acid or glutamic acid bead. Go-like AICG2+ potentials between histone cores and DNA, electrostatic interactions, and excluded volume effects were defined between DNA and proteins. Debye–Hückel-type electrostatic interactions were employed. Five Langevin dynamics simulations were performed at 300 K with a time unit of 0.2 with CafeMol, for a total of 5,000,000 steps. Since Pepsi-SAXS requires a fully atomistic model to calculate the scattering profile, CG-MD snapshots were reverse-mapped to atomistic models using BBQ, SCWRL4, and DNAbackmap[65–67].

**Reporting summary**. Further information on research design is available in the Nature Portfolio Reporting Summary linked to this article.

## Data availability

The MNase sequencing data have been deposited in the Gene Expression Omnibus with the accession (GEO) number GSE224789. The cryo-EM map and the atomic coordinates of the OLTN have been deposited in the Electron Microscopy Data Bank (EMDB) and the Protein Data Bank (PDB), under the accession codes EMD-35448 and PDB ID 8IHL, respectively. All data presented in this study are included in the main text or the supplementary information. Uncropped images are provided in Supplementary Fig. 6. Any additional information is available from the corresponding authors upon request.

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

## Acknowledgements

We are grateful to Y. Iikura, Y. Takeda, Y. Fukaya, and M. Dacher (University of Tokyo) for their assistance. We thank Dr. M. Kikkawa (University of Tokyo) for cryo-EM data collection. This work was supported in part by JSPS KAKENHI Grant Numbers JP22K06098 [to Y.T.], JP18H05534 [to H.K.], JP20H00449 [to H.K.], JP22J20665 [to T.F.], JP18H05527, JP20H00456 [to Y.O.], Research Support Project for Life Science and Drug Discovery (BINDS) from AMED under Grant Number JP22ama121009 [to H.K.], JP22ama121017j0001 [to Y.O.], JP22ama121002j001 [to M. Kikkawa, University of Tokyo], JST PRESTO Grant Number JPMJPR2026 [to K.Ma.], and JST ERATO Grant Number JPMJER1901 [to H.K.].

## Author contributions

M.N. and Y.K. reconstituted the OLTN, and M.N. H.T., K.N., and Y.T. performed the cryo-EM analysis. M.N., T.F., K.Ma., and Y.O. performed the genome-wide analysis with next-generation sequencing technology. K.Mo., M.Sh., and M. Su. performed the solution structural analysis and MD simulation analysis of the OLTN. H.K. designed the research and drafted the manuscript with M.N. All authors discussed and commented on the manuscript.

## Competing interests

The authors declare no competing interests.
