## [Peer Review File · Communications Biology]

Reviewers' comments:

Reviewer #1 (Remarks to the Author):

Evaluation of the manuscript "Formation and structure of the overlapping tri-nucleosome composed of hexasome-hexasome-octasome moieties" by Nishimura, Fujii et al.

In this manuscript, the authors report the existence of a chromatin structure found in human cells downstream of transcription start sites in the genome, corresponding to an overlapping tri-nucleosome composed of hexasome-hexasome-octasome moieties spanning a larger DNA fragment than those described so far. They have reconstituted this overlapping tri-nucleosome in vitro and obtained its cryo-EM structure at an intermediate resolution.

The work appears to be technically sound, the manuscript is well written and easy to follow. However, addressing several important points listed below would strengthen and clarify the manuscript. Moreover, certain terms in the title and abstract could be changed so as to better reflect the contents of this manuscript. In lines 28-30 of the abstract: "our genome-wide analysis [...] revealed [...] the overlapping tri-nucleosome [...]" the tone is very affirmative, yet the experimental data supporting this claim is rather thin. I would have preferred the more cautious tone that the authors keep in the whole manuscript: "the" should be replaced by "that could correspond to an" or additional data or discussion is needed to make it more convincing.

Similarly, the title "Formation and structure" is ambiguous. If the word "formation" is used to refer to the in vitro reconstitution, the title should be changed to "reconstitution". In the manuscript there is no evidence nor very much discussion about the formation of the overlapping tri-nucleosome. Although their proposed hypothesis regarding the formation of overlapping tri-nucleosome assemblies seems plausible, the authors should either provide additional experimental evidence supporting this hypothesis or a stronger discussion of this aspect, or change the title.

Additionally, the authors might want to consider the following remarks when revising the paper:

(1) In the analysis of MNase protection of chromatin isolated from HeLa cells, although the conclusions about the overlapping tri-nucleosome protecting the 300-350bp DNA fragment might be the most likely hypothesis, the authors mention line 93 (this is a good thing) the technical limitation. Have the authors ensured that there are no other proteins other than core histones in their sample? This could be achieved by analysis on a silver-stained SDS-PAGE gel or by nano-scale liquid chromatographic tandem mass spectrometry (nLC-MS/MS) that would additionally allow to know the relative quantity of each histone, which would guarantee the result regarding the hexasome-hexasome-octasome, instead of an octasome-octasome or hexasome-octasome bound to other proteins.

Providing additional experimental data to support their hypothesis and extending the discussion of the nature of the overlapping tri-nucleosome assembly by explaining that it is the most likely hypothesis given the known information about the octasome, hexasome, tetrasome, di-tetrasome and their DNA length coverage, propensity to appear, e.g. by relying on their reference 4 (Lavelle et al, 2007), would greatly reinforce this study.

(2) Line 110 : "The salt dialysis method". The authors should add an appropriate reference or explain the method.

(3) I would have liked the authors to explain and make more explicit in the manuscript why and how they proceed to construct the 125bp-103bp-125bp DNA (line 106-107). Did the authors try multiple DNA preparations, for example?

(4) Line 115-116 : Here the comparison between the results of the in vivo and in vitro MNase experiments is not clear because "300-350bp" refers to fig1,d where the number is 349bp (corresponding to the position more than the lengths if I am not mistaken) ; so the reference should be fig 1,e?

(5) Purification of the reconstituted overlapping tri-nucleosome and verification of the consistency of the structure with the results obtained on chromatin isolated from HeLa cells are important and well performed. Supplementary figure 2 could be incorporated into the main figures after figure 1.

(6) Related to (5), did the authors influence the reconstitution by adding less H2A and H2B protein, or did they mix stoichiometric amounts of each?

(7) The sample used for cryo-EM looks homogeneous. During GraFix procedure, are there any other objects resulting from the reconstitution? Could the authors comment on this aspect?

(8) Related to (7), In the 3D classifications in supplementary figure 3, one class (8,5%) has a different shape; the angle between two nucleosome moieties seems different from the others. Could the authors comment on this aspect?

(9) Line 120-121: Add the final resolution of the structure.

(10) Even if the resolution does not allow to explore certain details at the molecular level, the authors should make an effort to describe their structure in more detail and discuss it in light of earlier studies and other available structures of chromatin assemblies. For example, some ideas that could be explored:

- Could it be compared to the crystallographic structure of the overlapping di-nucleosome, which appears to be more compact than the presented structure, and to the overlapping di-nucleosome in solution especially in the most open and relaxed states (Matsumoto et al., 2020)?
- Since overlapping tri-nucleosome appears looser, does this mean that some interactions are lost compared to the overlapping di-nucleosome in its compact form?
- What about the interaction lost upon dimer removal? Do they appear to be compensated?
- Perhaps the electrostatic surface could provide information?
- What about the angle of the DNA between two moieties, which does not appear to be the same in the two overlapping structures? Could the DNA sequence matter?

(11) Could the authors then implement the discussion about the loosening between the octasome and hexasomes? Could this have a biological role? in terms of loosening the structure, could the empty space be compatible with the positioning of the missing dimer? if so, and because it is not present in the structure, could this be explained and placed in a nucleosome or chromatin context?

(12) Line 177 : What is the volume of HeLa nuclei suspension used?

Line 178 : MNase digestion is stopped by EDTA at what final concentration? and no proteinase K to digest remaining proteins? Is ethanol precipitation sufficient to dissociate bound proteins to be sure that are not still associated during the electrophoresis, which could delay the DNA migration and bias the migration?

(13) Line 197: "The OLTN was prepared with [...] and a purified histone octamer ". I would have liked to see how the histone octamer is purified even briefly if you refer to a previous study. Also, for the salt dialysis method, recall for (2). For a broad audience such as a Communications biology readers who might not be familiar with this kind of project, it might be easier to know how you proceed at

least in outline: purify and assemble the OLTN by salt dialysis.

(14) Line 235: What sample concentration was used to make the grids? Usually, for nucleosomes, the concentrations can be determined using the absorbance of DNA at 260nm. Could the authors specify the blotting conditions: temperature, force, time, and humidity in the blotting chamber?

In the cryo-EM data collection method: defocus range and the total electron dose should be added.

(15) The number of images (line 244) and the final resolution reached (line 254) should be added.

(16) There is no reference to Table 1 in the manuscript.

(17) Could the authors check or comment on the quality of the model given the number of residues that have an unusual fit to the map in the pdb validation report, compared to other structures with the same resolution?

Reviewer #2 (Remarks to the Author):

ATP-dependent chromatin remodelers move nucleosomes around, and this has been well-characterized. The open question is: what happens when thus moved nucleosomes 'bump into each other' and run out of linker DNA. This is addressed and potentially answered in this manuscript, from the authors who have previously identified and characterized an overlapping dinucleosome. The structure of an overlapping trinucleosome (determined by cryoEM at low resolution) is presented, consisting of two hexasomes and one octasome. This is very intriguing and an important finding.

Using similar approaches as in their previous publication, they confirmed the protection of 250 bp DNA corresponding to 'OLDN', and also identified larger protected DNA fragments in the same area. Particles on 350 bp (with judiciously placed 601 positioning elements) were prepared in vitro (but see question below) and upon crosslinking were analyzed by cryoEM. As such, these structures are quite interesting and also plausible, but overall I feel that a bit more depth in their description and analysis would be required. Perhaps most glaringly, the possibility that crosslinking artificially enhances a transient structure should be discussed / addressed. In-solution characterization of the OLTN is non-existent (apart from MNase). Additionally, the placement of the positioning elements likely predetermines the structures, and attempts should be made on 'normal' DNA to test whether similar structures are obtained. I appreciate that structure determination on random DNA might be difficult, but solution-state approaches (such as mass photometry or AUC) should be done to compare assemblies on 601 with different spacings perhaps, but more importantly on one or two DNA sequences where such OLTNs were found in cells.

Abstract:

1. should state the context in which the dinucleosome was found as a chromatin unit, otherwise misleading
2. downstream of the transcription start site – this is too general and somewhat misleading. all transcription start sites?

Introduction:

1. page 4, line 63: do the authors really mean to be so tentative about this peer-reviewed paper. fine if yes (and they don't believe the results), but if not please rephrase (note: this is not a paper from this reviewer's lab...).

Results

1. could the authors give an indication of how prevalent these structures (OLDN and OLTN) are with

respect to 'normally spaced' nucleosomes? it might be buried in the figure, but would be good to spell out.

2. Sup figure 2: it is surprising that the described procedures result in the exclusive formation of these rather contrived structures, and no other products are observed. I would be curious to see the results of the reconstitution prior to prep cell purification. How are the products obtained affected by the spacing of the positioning element (601?). How were the lengths in 2d obtained (should be in figure legend).

3. The OLTN structure was determined to intermediate resolution which is fine because the structure of the nucleosome components are well-determined. However, the structure was stabilized by GRAFIX (crosslinking) and this of course might artificially stabilize otherwise unstable (irrelevant) structures. As such, more work is required to characterize this structure in solution.

4. Fig. 3: the closeup of the interfaces isn't helpful – there seem to be virtually no contacts. Is this true? How flexible is the relative orientation of the individual hexasome / octasome moieties?
Discussion: Overall a bit short and rather superficial; shortcomings of the study (see above) are not discussed. The last paragraph is too speculative and lacks context. E.g. what does this mean? 'The OLDN and OLTN can be accommodated within the PIC without substantial steric clashes. Therefore, the OLDN and OLTN formed at the +1 position may function as regulators of the initiation of transcription elongation.' This seems very vague and unsubstantiated.

Reviewer #3 (Remarks to the Author):

Nishimura, Fujii et al. present the cryo-EM structure of an overlapping trinucleosome consisting of two hexasomes, and one intact octasome at 7.8 Å resolution. The authors performed MNase-seq and identified the presence of genomic fragments that show a protected footprint of ~350 bp at a position downstream of the transcription start site. The authors hypothesized that this fragment could represent an overlapping trinucleosome structure and reconstituted nucleosomes on a ~350 bp long DNA fragment. Using cryo-EM, they obtained a structure of the reconstituted nucleosome entity, revealing an overlapping trinucleosome. The authors argue that the overlapping trinucleosome could be generated through the action of chromatin remodelers. However, they do not provide any direct evidence for this claim.

The study is overall of sufficient technical quality and adds an additional aspect to the study of overlapping nucleosomes without clarifying or elucidating the mechanism of their generation, regulatory role, or function. The paper is overall very short and sparse. Important details regarding the cryo-EM processing are lacking.

Major concerns

1) The authors argue that solely chromatin remodeling action is responsible for the generation of the overlapping nucleosomes structures. However, they should also discuss in the introduction and discussion if transcription activity itself through retrograde movement of nucleosome during transcription as observed by the Farnung lab and by the Kurumizaka/Sekine labs could also lead to the generation of these overlapping nucleosome structures. Overall, the authors do not provide any evidence that overlapping trinucleosomal structures are generated by chromatin remodelers and do not sufficiently discuss other mechanisms that could lead to their creation. This should be clarified in the text.

2) It is unclear if state-of-the-art cryo-EM processing such as particle polishing and local CTF refinements were applied to the data set to obtain higher resolutions.

Minor concerns

- 1) The authors should provide the resolution for the FSC 0.143 and FSC 0.5 criteria in the FSC curves. A model-to-map FSC curve is missing.
- 2) It remains unclear why other 3D classes were omitted during further processing (e.g., the class with 16.9 % occupancy). This should be sufficiently described in the methods section.
- 3) Scale bar is missing for the 2D classes.
- 4) A figure showing representative densities with the built model is fully lacking.
- 5) The authors should report the Q score for their structure.
- 6) Resolutions throughout the paper should only be reported to one significant figure beyond the decimal point.

Reviewer #1:

Evaluation of the manuscript “Formation and structure of the overlapping tri-nucleosome composed of hexasome-hexasome-octasome moieties” by Nishimura, Fujii et al.

In this manuscript, the authors report the existence of a chromatin structure found in human cells downstream of transcription start sites in the genome, corresponding to an overlapping tri-nucleosome composed of hexasome-hexasome-octasome moieties spanning a larger DNA fragment than those described so far. They have reconstituted this overlapping tri-nucleosome in vitro and obtained its cryo-EM structure at an intermediate resolution.

The work appears to be technically sound, the manuscript is well written and easy to follow. However, addressing several important points listed below would strengthen and clarify the manuscript. Moreover, certain terms in the title and abstract could be changed so as to better reflect the contents of this manuscript. In lines 28-30 of the abstract: “our genome-wide analysis [...] revealed [...] the overlapping tri-nucleosome [...]” the tone is very affirmative, yet the experimental data supporting this claim is rather thin. I would have preferred the more cautious tone that the authors keep in the whole manuscript: “the” should be replaced by “that could correspond to an” or additional data or discussion is needed to make it more convincing.

Similarly, the title “Formation and structure “ is ambiguous. If the word “formation” is used to refer to the in vitro reconstitution, the title should be changed to “reconstitution “. In the manuscript there is no evidence nor very much discussion about the formation of the overlapping tri-nucleosome. Although their proposed hypothesis regarding the formation of overlapping tri-nucleosome assemblies seems plausible, the authors should either provide additional experimental evidence supporting this hypothesis or a stronger discussion of this aspect, or change the title.

Reply)

Thank you very much for this comment. We changed the title to “Genome-wide mapping and cryo-EM structural analyses of the overlapping tri-nucleosome composed of hexasome-hexasome-octasome moieties”, and modified the abstract in the revised manuscript.

Additionally, the authors might want to consider the following remarks when revising the paper:

(1) In the analysis of MNase protection of chromatin isolated from HeLa cells, although the conclusions about the overlapping tri-nucleosome protecting the 300-350bp DNA fragment might be the most likely hypothesis, the authors mention line 93 (this is a good thing) the technical limitation. Have the authors ensured that there are no other proteins other than core histones in their sample?

This good be achieved by analysis on a silver-stained SDS-PAGE gel or by nano-scale liquid chromatographic tandem mass spectrometry (LC-MS/MS) that would additionally allow to know the relative quantity of each histone, which would guarantee the result

regarding the hexasome-hexasome-octasome, instead of an octasome-octasome or hexasome-octasome bound to other proteins.

Providing additional experimental data to support their hypothesis and extending the discussion of the nature of the overlapping tri-nucleosome assembly by explaining that it is the most likely hypothesis given the known information about the octasome, hexasome, tetrasome, di-tetrasome and their DNA length coverage, propensity to appear, e.g by relying on their reference 4 (Lavelle et al, 2007), would greatly reinforce this study.

Reply)

Thank you very much for this comment. Our genome-wide mapping data have been obtained with purified genomic DNA fragments prepared by an extensive digestion with MNase. This allows the mapping of genomic loci with 300-350 bp DNA protection. However, the purification of the protein-DNA complexes containing these DNA fragments has not been successfully accomplished yet. Therefore, the proposed experiments are technically impossible at this point of time. As this reviewer pointed out, we cannot exclude the possibility that non-histone DNA-binding proteins are still bound to the DNA. We mentioned this in lines 105-109 of the revised manuscript.

(2) Line 110 : “The salt dialysis method”. The authors should add an appropriate reference or explain the method.

Reply)

We added the details of the salt dialysis method in the Methods section (pp. 10-11).

(3) I would have liked the authors to explain and make more explicit in the manuscript why and how they proceed to construct the 125bp-103bp-125bp DNA (line 106-107). Did the authors try multiple DNA preparation, for example?

Reply)

We explained the details of the DNA preparation in lines 115-119 and the Methods section (pp. 10-11).

(4) Line 115-116 : Here the comparison between the results of the in vivo and in vitro MNase experiments is not clear because “300-350bp” refers to fig1,d where the number is 349bp (corresponding to the position more than the lengths if I am not mistaken) ; so the reference should be fig 1,e?

Reply)

Thank you very much for this comment. We corrected it accordingly (line 127).

(5) Purification of the reconstituted overlapping tri-nucleosome and verification of the consistency of the structure with the results obtained on chromatin isolated from HeLa cells are important and well performed. Supplementary figure 2 could be incorporated into the main figures after figure 1.

Reply)

Thank you very much. We moved Supplementary Fig. 2 to the new Fig. 2.

(6) Related to (5), did the authors influence the reconstitution by adding less H2A and H2B protein, or did they mix stoichiometric amounts of each?

Reply)

We reconstituted the OLTN with the pre-reconstituted histone octamer. Therefore, the amounts of the histones are stoichiometric. We added this information in the Methods section "Reconstitution and purification of the OLTN" in the revised manuscript.

(7) The sample used for cryo-EM looks homogeneous. During GraFix procedure, are there any other objects resulting from the reconstitution? Could the authors comment on this aspect?

Reply)

As this reviewer pointed out, the reconstituted OLTN sample is quite homogenous during the GraFix procedure. We presented the gel image of the GraFix fractions of the OLTN in the new Supplementary Fig. 2.

(8) Related to (7), In the 3D classifications in supplementary figure 3, one class (8,5%) has a different shape; the angle between two nucleosome moieties seems different from the others. Could the authors comment on this aspect?

Reply)

The 3D map (8.5%) was obtained with a possibly different orientation between the OLTN part and the proximal hexasome portion, but this is uncertain due to the noisy background. We commented on this fact in lines 134-137 of the revised manuscript.

(9) Line 120-121: Add the final resolution of the structure.

Reply)

We added the resolution information of the cryo-EM structure in line 132 and lines 137-139 of the revised manuscript.

(10) Even if the resolution does not allow to explore certain details at the molecular level, the authors should make an effort to describe their structure in more detail and discuss it in light of earlier studies and other available structures of chromatin assemblies. For example, some ideas that could be explored:

- Could it be compared to the crystallographic structure of the overlapping di-nucleosome, which appears to be more compact than the presented structure, and to the overlapping di-nucleosome in solution especially in the most open and relaxed states (Matsumoto et al., 2020)?*
- Since overlapping tri-nucleosome appears looser, does this mean that some interactions are lost compared to the overlapping di-nucleosome in its compact form?*

- *What about the interaction lost upon dimer removal? Do they appear to be compensated?*
- *Perhaps the electrostatic surface could provide information?*
- *What about the angle of the DNA between two moieties, which does not appear to be the same in the two overlapping structures? Could the DNA sequence matter?*

Reply)

Thank you very much for these comments. In the revised manuscript, we described the structural comparison of the OLTN with the previous crystal and solution structures of the OLDN, in the last paragraph of the Discussion section. We also performed solution structural analyses of the OLTN by small angle X-ray scattering and coarse-grained molecular dynamics analyses. These new data support our conclusion that the cryo-EM structure of the OLTN reflects its solution structure. Given the dynamic arrangement of the hexasomal and octasomal portions in the OLTN, the histone-DNA and histone-histone interactions between the subnucleosomal portions may not be stable but transient around the cryo-EM structure. These results are described in the new Results section “Small angle X-ray and molecular dynamics simulation analyses of the OLTN in solution” with the new Fig. 5. In addition, to test the DNA sequence dependence of the OLTN formation, we performed the OLTN reconstitution with a native genomic DNA sequence, in which nucleosome remodeling is reportedly promoted by the SWI/SNF nucleosome remodeling complex. We then found that the OLTN can be formed with the native DNA sequence. These new data are presented in the new Supplementary Fig. 4, and are discussed in lines 140-148.

(11) Could the authors then implement the discussion about the loosening between the octasome and hexasomes? Could this have a biological role? in terms of loosening the structure, could the empty space be compatible with the positioning of the missing dimer? if so, and because it is not present in the structure, could this be explained and placed in a nucleosome or chromatin context?

Reply)

According to this reviewer’s suggestion, we implemented the discussion about the loosening of the OLTN. To do so, as described above, we performed the SAXS and MD analyses of the solution structure of the OLTN. We then elaborated the discussion about the loosening between octasome and hexasome. These new data are described in the new Results section “Small angle X-ray and molecular dynamics simulation analyses of the OLTN in solution” with the new Fig. 5. We also discussed the possible biological significance of the structural properties of the OLTN in the last paragraph of the Discussion section. The location of the missing H2A-H2B dimers is shown in Fig. 4b and presented as close-up views in Fig. 4c and d.

(12) Line 177 : What is the volume of HeLa nuclei suspension used?

Line 178 : MNase digestion is stopped by EDTA at what final concentration? and no proteinase K to digest remaining proteins? Is ethanol precipitation sufficient to dissociate bound proteins to be sure that are not still associated during the

electrophoresis, which could delay the DNA migration and bias the migration?

Reply)

We added the detailed protocol for the sample preparation of HeLa cell nuclei and MNase digestion in the Methods section (pp. 9-10).

(13) Line 197: "The OLTN was prepared with [...] and a purified histone octamer ". I would have liked to see how the histone octamer is purified even briefly if you refer to a previous study. Also, for the salt dialysis method, recall for (2). For a broad audience such as a Communications biology readers who might not be familiar with this kind of project, it might be easier to know how you proceed at least in outline: purify and assemble the OLTN by salt dialysis.

Reply)

As this reviewer requested, we described the details of the histone octamer purification and the reconstitution of the OLTN in the Methods section (pp. 10-11).

(14) Line 235: What sample concentration was used to make the grids? Usually, for nucleosomes, the concentrations can be determined using the absorbance of DNA at 260nm. Could the authors specify the blotting conditions: temperature, force, time, and humidity in the blotting chamber?

In the cryo-EM data collection method: defocus range and the total electron dose should be added.

Reply)

We added detailed information about the sample preparation for the cryo-EM analysis and data collection in the Methods sections "Preparation of the OLTN for cryo-EM analysis" and "Cryo-EM data collection" (pp. 12-14).

(15) The number of images (line 244) and the final resolution reached (line 254) should be added.

Reply)

We added the number of images and the final resolution in the Methods section "Image processing" (p.14).

(16) There is no reference to Table 1 in the manuscript.

Reply)

We added the reference to Supplementary Table 1 in the revised manuscript.

(17) Could the authors check or comment on the quality of the model given the number of residues that have an unusual fit to the map in the pdb validation report, compared to other structures with the same resolution?

Reply)

We improved the model quality and deposited the new pdb model of the OLTN to the wwPDB. We uploaded the new pdb validation report with the revised manuscript. We also replaced all models in the revised manuscript.

Reviewer #2 (Remarks to the Author):

ATP-dependent chromatin remodelers move nucleosomes around, and this has been well-characterized. The open question is: what happens when thus moved nucleosomes 'bump into each other' and run out of linker DNA. This is addressed and potentially answered in this manuscript, from the authors who have previously identified and characterized an overlapping dinucleosome. The structure of an overlapping trinucleosome (determined by cryoEM at low resolution) is presented, consisting of two hexasomes and one octasome. This is very intriguing and an important finding.

Using similar approaches as in their previous publication, they confirmed the protection of 250 bp DNA corresponding to 'OLDN', and also identified larger protected DNA fragments in the same area. Particles on 350 bp (with judiciously placed 601 positioning elements) were prepared in vitro (but see question below) and upon crosslinking were analyzed by cryoEM. As such, these structures are quite interesting and also plausible, but overall I feel that a bit more depth in their description and analysis would be required. Perhaps most glaringly, the possibility that crosslinking artificially enhances a transient structure should be discussed / addressed. In-solution characterization of the OLTN is non-existent (apart from MNase). Additionally, the placement of the positioning elements likely predetermines the structures, and attempts should be made on 'normal' DNA to test whether similar structures are obtained. I appreciate that structure determination on random DNA might be difficult, but solution-state approaches (such as mass photometry or AUC) should be done to compare assemblies on 601 with different spacings perhaps, but more importantly on one or two DNA sequences where such OLTNs were found in cells.

Reply)

Thank you very much for these insightful comments. According to this reviewer's concern, in the revised manuscript, we performed the SAXS analysis of the OLTN without crosslinking in solution. We then found that the solution structure of the OLTN is perfectly consistent with the cryo-EM structure. In addition, we performed the coarse-grained molecular dynamics analysis of the OLTN. These new data support our conclusion that the cryo-EM structure of the OLTN reflects its solution structure. These results are described in the new Results section "Small angle X-ray and molecular dynamics simulation analyses of the OLTN in solution" with the new Fig. 5. As requested, we tried to reconstruct the OLTN using a native DNA sequence instead of a Widom601 DNA sequence, and successfully obtained the 2D class averages of the OLTN containing a native DNA sequence. These new data related to the OLTN formation with a native genomic DNA sequence are now presented in the new Supplementary Fig. 4, and the results are described in lines 140-147.

Abstract:

1. should state the context in which the dinucleosome was found as a chromatin unit, otherwise misleading

Reply)

We rewrote the corresponding sentence as “The overlapping di-nucleosome has been proposed as a product of chromatin remodeling around the transcription start site, and previously found as a chromatin unit, in which about 250 base pairs of DNA continuously bind to the histone core composed of a hexamer and an octamer”.

2. downstream of the transcription start site – this is too general and somewhat misleading. all transcription start sites?

Reply)

We rewrote the corresponding sentence to be more specific as “In the present study, our genome-wide analysis of human cells suggested another higher nucleosome stacking structure, the overlapping tri-nucleosome, which wraps about 300-350 base-pairs of DNA in the region downstream of certain transcription start sites of actively transcribed genes”.

Introduction:

1. page 4, line 63: do the authors really mean to be so tentative about this peer-reviewed paper. fine if yes (and they don't believe the results), but if not please rephrase (note: this is not a paper from this reviewer's lab...).

Reply)

According to this reviewer's comment, in the revised manuscript, we removed the corresponding sentence from the Introduction section.

Results

1. could the authors give an indication of how prevalent these structures (OLDN and OLTN) are with respect to 'normally spaced' nucleosomes? it might be buried in the figure, but would be good to spell out.

Reply)

In the current method, the possible existence of the OLDN and OLTN can be deduced in cells; however, their prevalence relative to the normal nucleosome may be difficult to evaluate. Therefore, the rates of OLDN and OLTN formation relative to the nucleosome may be an important future issue to solve. We mentioned this in lines 194-198.

2. Sup figure 2: it is surprising that the described procedures result in the exclusive formation of these rather contrived structures, and no other products are observed. I would be curious to see the results of the reconstitution prior to prep cell purification. How are the products obtained affected by the spacing of the positioning element (601?).

Reply)

According to this reviewer's request, in the revised manuscript, we presented the non-denaturing gel fractionated by GraFix with the input sample before fractionation in the new Supplementary Fig. 2. We collected the fractions containing the homogeneous OLTN sample and prepared it for the cryo-EM analysis.

How were the lengths in 2d obtained (should be in figure legend).

Reply)

We explained the lengths in 2d in the legend for the new Fig. 2.

3. The OLTN structure was determined to intermediate resolution which is fine because the structure of the nucleosome components are well-determined. However, the structure was stabilized by GRAFIX (crosslinking) and this of course might artificially stabilize otherwise unstable (irrelevant) structures. As such, more work is required to characterize this structure in solution.

Reply)

Thank you very much for this suggestion. In the revised manuscript, we performed the SAXS analysis of the OLTN without crosslinking in solution. We then found that the solution structure of the OLTN is perfectly consistent with the cryo-EM structure. In addition, we performed the coarse-grained molecular dynamics analysis of the OLTN. These new data support our conclusion that the cryo-EM structure of the OLTN reflects its solution structure. These results are described in the new Results section "Small angle X-ray and molecular dynamics simulation analyses of the OLTN in solution" with the new Fig. 5.

4. Fig. 3: the closeup of the interfaces isnt helpful – there seem to be virtually no contacts. Is this true? How flexible is the relative orientation of the individual hexasome / octasome moieties?

Reply)

As this reviewer commented, the direct interactions between octasome and hexasome or hexasome and hexasome are not obvious in the current cryo-EM structure of the OLTN. In the revised manuscript, we described the structural arrangements between the subnucleosomal moieties in the new Results section "The OLTN structure". To discuss this issue, we also described the structural comparison of the OLTN with previous crystal and solution structures of the OLDN in the last paragraph of the Discussion section.

Discussion: Overall a bit short and rather superficial; shortcomings of the study (see above) are not discussed. The last paragraph is too speculative and lacks context. E.g. what does this mean? 'The OLDN and OLTN can be accommodated within the PIC without substantial steric clashes. Therefore, the OLDN and OLTN formed at the +1 position may function as regulators of the initiation of transcription elongation.' This

seems very vague and unsubstantiated.

Reply)

We further elaborated the discussion and modified the last paragraph in the revised manuscript.

Reviewer #3 (Remarks to the Author):

Nishimura, Fujii et al. present the cryo-EM structure of an overlapping trinucleosome consisting of two hexasomes, and one intact octasome at 7.8 Å resolution. The authors performed MNase-seq and identified the presence of genomic fragments that show a protected footprint of ~350 bp at a position downstream of the transcription start site. The authors hypothesized that this fragment could represent an overlapping trinucleosome structure and reconstituted nucleosomes on a ~350 bp long DNA fragment. Using cryo-EM, they obtained a structure of the reconstituted nucleosome entity, revealing an overlapping trinucleosome. The authors argue that the overlapping trinucleosome could be generated through the action of chromatin remodelers. However, they do not provide any direct evidence for this claim.

The study is overall of sufficient technical quality and adds an additional aspect to the study of overlapping nucleosomes without clarifying or elucidating the mechanism of their generation, regulatory role, or function. The paper is overall very short and sparse. Important details regarding the cryo-EM processing are lacking.

Reply)

Thank you very much for your comment. In the revised manuscript, we included additional experimentation, elaborating the results and discussion, and the details of the cryo-EM processing, as described below.

Major concerns

1) The authors argue that solely chromatin remodeling action is responsible for the generation of the overlapping nucleosomes structures. However, they should also discuss in the introduction and discussion if transcription activity itself through retrograde movement of nucleosome during transcription as observed by the Farnung lab and by the Kurumizaka/Sekine labs could also lead to the generation of these overlapping nucleosome structures. Overall, the authors do not provide any evidence that overlapping trinucleosomal structures are generated by chromatin remodelers and do not sufficiently discuss other mechanisms that could lead to their creation. This should be clarified in the text.

Reply)

Thank you very much. As this reviewer pointed out, it is also possible that the OLTN formation may occur by the transcription by RNAPII. In the revised manuscript, we discussed this possibility in the new second paragraph of the Discussion section.

2) It is unclear if state-of-the-art cryo-EM processing such as particle polishing and local CTF refinements were applied to the data set to obtain higher resolutions.

Reply)

We have presented the cryo-EM structure of the OLTN at 7.6 Å without performing Bayesian polishing and CTF refinements, because they didn't improve the overall resolution significantly.

Minor concerns

1) The authors should provide the resolution for the FSC 0.143 and FSC 0.5 criteria in the FSC curves. A model-to-map FSC curve is missing.

Reply)

We added the resolutions for both the FSC 0.143 and FSC 0.5 criteria in the FSC curves, and a model-to-map FSC curve in the new Supplementary Fig. 3.

2) It remains unclear why other 3D classes were omitted during further processing (e.g., the class with 16.9 % occupancy). This should be sufficiently described in the methods section.

Reply)

We improved our description regarding the 3D classification of the OLTN in the Methods section of the revised manuscript.

3) Scale bar is missing for the 2D classes.

Reply)

We added the scale bar for the 2D class averages in the new Supplementary Fig. 3 and the new Supplementary Fig. 4.

4) A figure showing representative densities with the built model is fully lacking.

Reply)

We added the built model of the OLTN docking into the cryo-EM map in the new Fig. 3.

5) The authors should report the Q score for their structure.

Reply)

We added the Q score in the new Supplementary Table 1.

6) Resolutions throughout the paper should only be reported to one significant figure beyond the decimal point.

Reply)

We changed the resolution to one significant figure beyond the decimal point in the revised manuscript and the new Supplementary Fig. 3.

REVIEWERS' COMMENTS:

Reviewer #1 (Remarks to the Author):

The authors have addressed in a satisfactory way most of my previous concerns, either by enriching the results and discussion of the structure obtained as well as discussing the results in a physiological context, or by providing a solid set of additional data. Indeed, they have carried out structural analyses of the OLTN solution by small-angle X-ray scattering and coarse-grained molecular dynamics analyses. I also appreciated that in order to perform 2D cryoEM classifications, the authors provided a new sample with a native sequence of DNA.

I congratulate the authors for all the additional work.

Regarding my previous point (1): I understand the difficulty of performing additional experiments in this period of time. I also recognize the effort made in the text regarding this point, but the added text doesn't bring additional information and becomes redundant with a previous sentence in the same paragraph. Therefore, I'm not sure that the comment added by the authors in lines 105-108 is necessary and this part of the text could be left as it was in the first version of the manuscript.

Regarding my previous comment (7) about the homogeneity of the cryo-EM sample after GraFix procedure and if they are other species resulting of the reconstitution:

With the addition of the new supplementary figure 2 is fine, the authors' response is fine, but the authors haven't commented on the other bands. Do they know what these other objects are?

There are still a few comments and modifications to be considered by the authors:

- At the end of the introduction, when the authors describe the results obtained (lines 74-78), they can now add in the revised manuscript the additional results obtained with their additional experiments.
- Line 65 "H2AH2B" dimer should be H2A-H2B.
- Line 94 "DNAbinding" should be DNA-binding.
- Line 119 and in some figures the H3 histone is specified to be H3.1 whereas in material and methods of these experiments is written H3. Authors have to be consistent.
- Line 123, the reference to supplementary figure 2 is not correct if I am not mistaken. Were the OLTNs cross-linked on a grafix gradient for MNase digestion? If so, the reference is correct, otherwise it has to be deleted.
- Line 129 reference to supplementary figure 2 should be added here.
- Line 151 "hexasomehexasome-octasome" should be hexasome-hexasome-octasome.
- Line 168: Authors should add that is Widom DNA to be more specific and avoid ambiguities due to the addition of new DNA for OLTN reconstitution for cryoEM.
- Line 360: shouldn't "Protease K" be proteinase K?

Reviewer #2 (Remarks to the Author):

The authors did an outstanding job in responding (and ultimately obliterating) most of my concerns. The SAXS experiments are particularly powerful, as are the recons on a 'random' ('native') DNA sequence. The added details in experimental section provide clarity, and the more careful wording provides the right context for the really intriguing results. This is now a very strong manuscript that will stand the test of time - congratulations to all authors [Karolin Luger]

Reviewer #3 (Remarks to the Author):

The authors have addressed all concerns that this reviewer has raised.

Reviewer #1 (Remarks to the Author):

General comment)

The authors have addressed in a satisfactory way most of my previous concerns, either by enriching the results and discussion of the structure obtained as well as discussing the results in a physiological context, or by providing a solid set of additional data. Indeed, they have carried out structural analyses of the OLTN solution by small-angle X-ray scattering and coarse-grained molecular dynamics analyses. I also appreciated that in order to perform 2D cryoEM classifications, the authors provided a new sample with a native sequence of DNA.

I congratulate the authors for all the additional work.

Reply)

Thank you very much for such a favorable comment. I revised the manuscript according to this reviewer's suggestion, as below.

Comment 1)

Regarding my previous point (1): I understand the difficulty of performing additional experiments in this period of time. I also recognize the effort made in the text regarding this point, but the added text doesn't bring additional information and becomes redundant with a previous sentence in the same paragraph. Therefore, I'm not sure that the comment added by the authors in lines 105-108 is necessary and this part of the text could be left as it was in the first version of the manuscript.

Reply)

Thank you very much. According to this suggestion, the corresponding sentences are replaced by the sentences in the first version of the manuscript.

Comment 2)

Regarding my previous comment (7) about the homogeneity of the cryo-EM sample after GraFix procedure and if they are other species resulting of the reconstitution:

With the addition of the new supplementary figure 2 is fine, the authors'

response is fine, but the authors haven't commented on the other bands. Do they know what these other objects are?

Reply)

The additional bands just after the OLTN reconstitution may be the OLDN (migrating faster than the OLTN), the OLTN with improper additional histone binding (migrating slower than the OLTN), and the OLTN with the different positioning of subnucleosome moieties. These possibilities are described in the text (ll. 118-122).

Additional comments)

There are still a few comments and modifications to be considered by the authors:

Comment)

- At the end of the introduction, when the authors describe the results obtained (lines 74-78), they can now add in the revised manuscript the additional results obtained with their additional experiments.

Reply)

In the revised manuscript, we added the description about additional results obtained by the revised experiments in the last paragraph of the Introduction.

Comment)

- Line 65 "H2AH2B" dimer should be H2A-H2B.

Reply)

We corrected it accordingly.

Comment)

- Line 94 "DNAbinding" should be DNA-binding.

Reply)

We corrected it accordingly.

Comment)

- Line 119 and in some figures the H3 histone is specified to be H3.1 whereas in material and methods of these experiments is written H3. Authors have to be consistent.

Reply)

We corrected H3.1 to H3, and explained that the H3 used in this work should be H3.1 in the Materials and Methods.

Comment)

- Line 123, the reference to supplementary figure 2 is not correct if I am not mistaken. Were the OLTNs cross-linked on a grafix gradient for MNase digestion? If so, the reference is correct, otherwise it has to be deleted.

Reply)

Thank you very much. As this reviewer noticed, this is mistakenly referenced. In the revised manuscript, we deleted this reference.

Comment)

- Line 129 reference to supplementary figure 2 should be added here.

Reply)

We corrected it accordingly.

Comment)

- Line 151 "hexasomehexasome-octasome" should be hexasome-hexasome-octasome.

Reply)

We corrected it accordingly.

Comment)

- Line 168: Authors should add that is Widom DNA to be more specific and avoid ambiguities due to the addition of new DNA for OLTN reconstitution for cryoEM.

Reply)

We specify the Widom 601 sequence for the OLTN sample used in the SAXS experiments.

Comment)

- Line 360: shouldn't "Protease K" be proteinase K?

Reply)

We corrected it accordingly.